# POOLING IMAGE DATASETS WITH MULTIPLE COVARIATE SHIFT AND IMBALANCE

**Sotirios Panagiotis Chytas**[*]
UW-Madison

**Vishnu Suresh Lokhande**
UW-Madison

**Vikas Singh**
UW-Madison

## ABSTRACT

Small sample sizes are common in many disciplines, which necessitates pooling roughly similar datasets across multiple institutions to study weak but relevant associations between images and disease outcomes. Such data often manifest shift/imbalance in covariates (i.e., secondary non-imaging data). Controlling for such nuisance variables is common within standard statistical analysis, but the ideas do not directly apply to overparameterized models. Consequently, recent work has shown how strategies from invariant representation learning provides a meaningful starting point, but the current repertoire of methods is limited to accounting for shifts/imbalances in just a couple of covariates at a time. In this paper, we show how viewing this problem from the perspective of Category theory provides a simple and effective solution that completely avoids elaborate multi-stage training pipelines that would otherwise be needed. We show the effectiveness of this approach via extensive experiments on real datasets. Further, we discuss how this style of formulation offers a unified perspective on at least 5+ distinct problem settings, from self-supervised learning to matching problems in 3D reconstruction. The code is available at https://github.com/SPChytas/CatHarm.

## 1 INTRODUCTION

Sample sizes of medical imaging datasets at a single institution are often small due to many reasons. Acquiring thousands of images can be infeasible due to budget/logistics. Also, if the scope of a study is narrow, only some individuals may be eligible due to inclusion criteria (e.g., have a specific genetic risk). To improve the statistical signal in retrospective analyses of existing datasets, one option is to pool similar data across multiple sites (Thompson et al., 2014). This offers a chance at discovering a real statistical effect, undetectable with small sample sizes.

Pooling data from multiple sites is common. A mature body of statistical literature (e.g., covariate matching (Stuart, 2010; Kim & Steiner, 2016), meta-analysis (Thompson et al., 2014; Rücker et al., 2021)) describes best practices, and mechanisms to account for the effect of a nuisance variable on the response (or target label) are well known. This allows obtaining associations between relevant predictors and the response/dependent variable of interest. For example, a standard analysis workflow may use *only* the images to predict a label (e.g., cognition) while controlling for nuisance variables such as race, gender or scanner type (Penny et al., 2007).

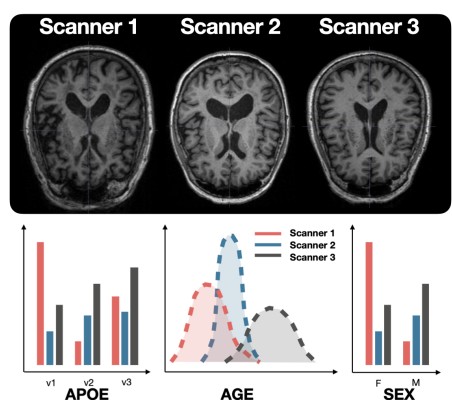

Figure 1: **Problem overview:** When pooling image datasets from different sources, there may be differences in the distribution of covariates. Covariates are secondary data for each individual that influences the images systematically. Top row shows MR images (say, different scanners). Second row shows how the covariate distributions (genetic risk, age or gender) varies across scanners (but have a shared support). Learning representations from a pooled dataset in a manner such that covariate variations are accounted for, is challenging.

---

[*]Corresponding author. Email: chytas@wisc.edu

**Controlling for covariates.** Deep neural network (DNN) models are now ubiquitous in medical image analysis (Ronneberger et al., 2015; Carin & Pencina, 2018). But when using such models, systematic mechanisms to deal with nuisance variables (or covariates), a key concern in data pooling, are still under development. Consider pooling MR images from two different sites where participant demographics distributions (e.g., age, sex etc) across sites are similar. If the scanners at the sites are different, we can include "scanner" or other continuous variables as a nuisance, and control for it within a general linear model. Features in the image $X$ deemed informative will explain variability in the response $Y$ *after* the variability due to "age" and "site" has been accounted for. In other words, the image data (and not the site-specific artifacts) should predict the response variable. Harmonization/pooling of imaging data in a way that accounts for (or removes) the influence of non-imaging covariates is often limited to shallow pre-processing using additive/multiplicative batch corrections, e.g., in Combat (Johnson et al., 2007; Fortin et al., 2018).

**Data pooling/harmonization and invariance.** In the context of representation learning, several results have identified the link between data pooling/harmonization and invariant representation learning or domain adaptation (Moyer et al., 2018; 2020), which have been utilized for downstream tasks (Lokhande et al., 2020). Initial approaches focused on image normalization techniques (e.g., histogram matching Nyúl & Udupa (1999)), which allowed controlling some variations in intensities across scanners or protocols. However, such approaches cannot systematically handle covariates. Ideas based on invariant representations (Bloem-Reddy & Teh, 2020; Li et al., 2014; Arjovsky et al., 2019) have allowed controlling for up to two covariates during representation learning. For example, one may ask that the model avoid using image features associated with two specific covariates (or in fairness terminology, sensitive attributes): age and site. Nonetheless, models that can easily remove (or control for) the influence of multiple covariates remain limited and so, either Combat-based (Johnson et al., 2007) pre-processing is used (Fortin et al., 2018) or CycleGAN-based schemes (Zhang et al., 2018; Nguyen et al., 2018) can harmonize the image data relative to a specific categorical variable (say, scanner). This problem is also relevant in the longitudinal setting Sauty & Durrleman (2022).

**The problem.** Different scanners can introduce systematic differences in the scans of the same person (Liu et al., 2020) and mitigation strategies continue to be a topic of recent work (Zhang et al., 2023). When there is some shift/disbalance in covariates (participant data including age, sex, and so on, which influence the scans) across sites and shared support is partial ("age" distribution in Fig. 1), the harmonization task becomes more involved. The goal is to learn representations as if the covariates were matched across the sites to begin with, see Fig. 1. Note that in contrast to covariate shift methods (Bickel et al., 2009), the shift here is not between train and test distributions, rather between different sources of data in the training set itself.

One recent attempt in (Lokhande et al., 2022) to learn deep representations from pooled data which can also handle (shift+imbalance) in covariates needs a two stage pipeline. The first stage obtains latent representations of the scan which is *equivariant* to age. The second stage enforces *invariance* to scanner and trained after freezing the first stage. Two covariates can be handled but an extension to *many* continuous (and categorical) covariates will need a complicated multi-stage model.

**Main Ideas.** In most learning tasks involving an equivariance or invariance criteria, the goal is to allow the model to benefit from structure and symmetry – in the data, the parameter space, or both. Frequently, one formalizes these ideas using group theory seeking invariance/equivariance to the action of the group (e.g., $\mathbb{SO}(n)$, $\mathbb{SE}(n)$, $\mathbb{S}_n$) (Bloem-Reddy & Teh, 2020; Kondor & Trivedi, 2018). More general criteria (beyond properties native to the chosen group) requires specialized treatment. Our **starting point** is based on the observation that Category theory (Fong & Spivak, 2019; MacLane, 2014) provides a rich set of tools by which the "structure" (either among the covariates or the data more generally) can be expressed easily. It is known that equivariance, invariance and many group-theoretic constructs will emerge as special cases because category theory provides a more abstract treatment. Once our criteria are expressed in this way, during training, the necessary constraints on the latent space fall out directly, and the formulation gracefully handles heterogeneity in many continuous/categorical covariates. No ad-hoc adjustments are needed.

**Contributions. (a)** On the **technical** side, we provide a general framework for imposing structure on the latent space by applying ideas from Category theory. Equivariant (and invariant) representation learning emerge as special cases. Further, this style of formalism unifies different formulations in vision and machine learning, under the same umbrella. **(b)** On the **practical** side, we strictly

generalize existing formulations to pool/harmonize multi-site datasets. While the existing two-stage formulation can deal with one categorical and one continuous covariate, our formulation places no restriction at all on the number of covariates. We show how the same model can also be used to heuristically approach certain hypothetical "what if" questions and offers competitive performance on public brain imaging datasets, with strictly more functionality/flexibility.

## 2 A BRIEF REVIEW OF CATEGORY THEORY

Category theory offers a way to study abstract "structures" (Eilenberg & MacLane, 1945; MacLane, 2014). A **Category** consists of two components **(a) Objects** that correspond to individual entities (e.g., scans/images), and **(b) Morphisms**, identified by the paths (or "arrows") between the objects. Each Object has an identity Morphism (a self-loop is often omitted when drawing the diagram).

**Remark 1** *The composition of two Morphisms $f : S_1 \to S_2$ and $g : S_2 \to S_3$ is a well-defined Morphism and it is denoted as $g \circ f : S_1 \to S_3$ (we should read it as g after f).*

**Example 2** *Consider the Category of Sets, in which the Objects are sets and the Morphisms correspond to functions (or matchings) between sets. Since $\exists f : S_1 \to S_2$, $g : S_2 \to S_3$ then $\exists h = g \circ f : S_1 \to S_3$.*

*In the Category of Sets $id_S : S \to S$ corresponds to the identity function: $id_S(s) = s, \; \forall s \in S$.*

**Definition 3** *Functors $F : \mathcal{S} \to \mathcal{T}$ give a relationship between the Objects and the Morphisms of two different Categories: source Category $\mathcal{S}$ and target Category $\mathcal{T}$. The conditions below hold,*

**(i)** $F(id_S) = id_{F(S)} \quad \forall S \in \mathcal{S}$

**(ii)** $F(g \circ f) = F(g) \circ F(f)$
$\forall f : S_1 \to S_2, g : S_2 \to S_3, \text{ in } \mathcal{S}$

**Remark 4** *The reader will see why the specification above is general, but useful. The functor $F$ takes us from the first Category to the next in a way that the structure of the source Category (objects and arrows) is fully preserved in the target Category because of the two conditions above.*

## 3 ENFORCING SYMMETRY AND STRUCTURE

Symmetry/structure in the data or parameters are key properties we exploit in learning tasks, e.g., invariance in features (Lowe, 1999), compressive sensing (Candes & Wakin, 2008) as well as DNN training (Sabour et al., 2017; Hinton, 2021). Encoding symmetry/structure can involve ideas as simple as rotations and cropping for data augmentation (Shorten & Khoshgoftaar, 2019; van Dyk & Meng, 2001; Chen et al., 2020b) to more sophisticated concepts (Cohen et al., 2018; Fuchs et al.,

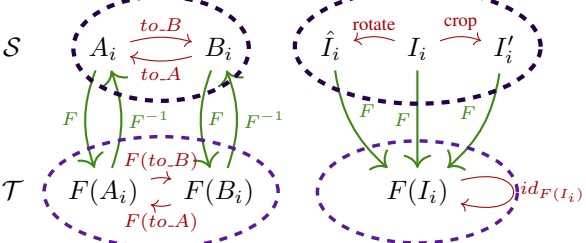

Figure 2: A Category theoretic view of CycleGAN (left) and SimCLR (right)

2020). This criteria can also be specified on the latent space, e.g., via geometric/topological priors (Moor et al., 2020; Ghosh et al., 2020) or other conditions (Chen et al., 2019; Oring et al., 2021). From Fig. 1, if there is a natural structure (say, in covariate space), then the structure of the representations learned on the latent space should respect that. To setup this description, we first discuss how this perspective can relate distinct ideas under an abstract (but simple) formalism.

### 3.1 REINTERPRETING CYCLEGAN USING CATEGORY THEORY

Image translation uses a set of image pairs and trains a DNN model to map between them. CycleGAN (Zhu et al., 2017) is a popular framework for image translation that removes the need for

paired image pairs in the training data, and learns the map from one domain (or class) to the other. Denoting image domains as *type A* and *type B*, the goal in CycleGAN is to learn how to map a *type A* image to a *type B* image and back; which we can denote as *to_B* and *to_A*: type $A$ (and type $B$) can mean images of horses (and zebras). We will avoid defining a category for the distributions over the images for simplicity, and so a training dataset made up of paired images will suffice to convey the key idea. So, for every image of type $A$, there is a similar image of type $B$. We want to learn a mapping (Functor) to a latent space that preserves the action of change from type $A$ to $B$ and back. This is shown in Fig. 2 (left), from which we can directly read off the following constraints,

1. $F$ is a fully-faithful Functor meaning: $\exists F^{-1} : \mathcal{T} \to \mathcal{S}$ such that $F^{-1} \circ F(s) = id_s$
2. Composition: to_B $\circ$ to_A$(B_i) = id_{B_i}(B_i) = B_i$    to_A $\circ$ to_B$(A_i) = id_{A_i}(A_i) = A_i$

The first pair of constraints is implicit in autoencoder-like models. The second pair of constraints define the cycle consistency conditions in CycleGAN. We note that a different formalism for CycleGAN was described in (Gavranović , 2020) by considering CycleGAN as a schema where cycle-consistencies enforce composition invariants in that Category.

## 3.2 REINTERPRETING SIMCLR USING CATEGORY THEORY

SimCLR (Chen et al., 2020b) is a *self-supervised learning* method where one attempts to map the images to a latent space which is invariant to transformations such as rotations, croppings, and so on. This can be instantiated using an invariant Functor (see Fig. 2). Unlike CycleGAN, such an invariant Functor $F$, is no longer fully-faithful (i.e., $\nexists F^{-1}$ such that $F \circ F^{-1} = id_x$).

## 3.3 REINTERPRETING OTHER FORMULATIONS USING CATEGORY THEORY

**(a) Latent space interpolation.** A number of approaches seek to perform interpolation in latent space (Chen et al., 2019; Oring et al., 2021)), informed by "covariates" associated with the original images (e.g., age, sex, hair length). The recipe described in the two examples above lends itself directly to describing such formulations, and will be subsumed by our formulation in the next section. **(b) Rotation and Permutation synchronization.** Rotation synchronization is a central problem in Cryo-EM image reconstruction (Marshall et al., 2022; Bendory et al., 2022; Tagare et al., 2008). In computer vision, permutation synchronization Pachauri et al. (2013) seeks to find replicable matches of keypoints across many images of the same 3D scene, e.g., for 3D reconstruction from datasets such as PhotoTourism (Li et al., 2022; Birdal et al., 2021; Birdal & Şimşekli, 2019) as well as in robotics (Leonardos et al., 2017). Since the synchronization task always involves consistency constraints over the symmetric Group, and Groups are a Category, the reinterpretation is immediate. **(c) Equivariance and Invariance.** Many problems in learning benefit from equivariance and invariance. Recall that equivariance is formally defined in terms of an action of a Group $G$.

**Definition 5 (Equivariance)** *A mapping $f : \mathcal{S} \to \mathcal{T}$ defined over measurable Borel spaces $\mathcal{S}$ and $\mathcal{T}$ is said to be $G$-equivariant under the action of group $G$ iff*

$$f(g \cdot s) = g \cdot f(s), \quad g \in G \tag{1}$$

A more general definition of equivariance follows directly from the Functor's definition.

**Definition 6** *A Functor $F : \mathcal{S} \to \mathcal{T}$ is defined as a mapping from $\mathcal{S}$ to $\mathcal{T}$ such that*

$$F(id_S) = id_{F(S)} \quad \forall S \in \mathcal{S} \tag{2}$$

$$F\big(g(S_1)\big) = F(g)\big(F(S_1)\big), \quad \forall s, g : S_1 \to S_2 \in \mathcal{S} \tag{3}$$

If $\mathcal{S}, \mathcal{T}$ are Borel spaces, and $g, F(g)$ belong to a Group $G$, then we obtain the group-theoretic equivariance definition. Category theory gives a more general result with no restrictions on the form of the relationships (or Morphisms) $g$. Similarly, invariance (e.g., in Group theory) is defined as

**Definition 7** *A mapping $f : \mathcal{S} \to \mathcal{T}$ defined over measurable Borel spaces $\mathcal{S}$ and $\mathcal{T}$ is said to be $G$-invariant under the action of group $G$ iff*

$$f(g \cdot s) = f(s), \quad g \in G \tag{4}$$

In Category theory, this results in a special type of Functor:

**Definition 8 (Invariance)** *A Functor $F : \mathcal{S} \to \mathcal{T}$ is invariant to the Morphism $g : S_1 \to S_2$ if $F(g) = id_{F(S)}$.*

### 3.4 A SIMPLE SANITY CHECK OF BENEFITS ON MNIST DATASET

Consider the case where the objects of the source Category $\mathcal{S}$ consist of MNIST images (LeCun et al., 1998). Since these images represent integers on the number line, we can ask whether we can learn a latent space which allows algebraic manipulations, with one (or more) basic operations defined on it. If our operation of interest was "addition", a primitive we will need for counting would be the "$+1$" operation. To keep things simple, we will focus on this operation – where the source Category $\mathcal{S}$ consists of Morphisms that represent the "$+1$" difference between two subsequent digits. It is easy to check that we have indeed defined a Category since each Object has an identity Morphism and the Morphisms compose. The target Category $\mathcal{T}$ (i.e., our latent space) can consist of vectors $v \in \mathbb{R}^n$ as objects. Orthogonal linear mappings $W \in \mathbb{R}^{n \times n}$ as Morphisms will suffice. In our pooling/harmonization problem, the morphisms will reflect traversing the axis of each covariate.

For this MNIST example, our goal is to impose the "$+1$" operation in the latent space, see Fig. 3. We define our loss to express precisely the equivalence for subsequent digits as well as a subset of their $+1$ compositions that can be read off of Fig. 3. The model is *not* presented data for "$-1$" operations but discovers it due to the special structure of $W$ (orthogonal, so $W^{-1}$ exists). Then, after training, when presented a "seed" image of 6, the model can (forward/backward) traverse the latent space: a query "$6 - 4$" starts from

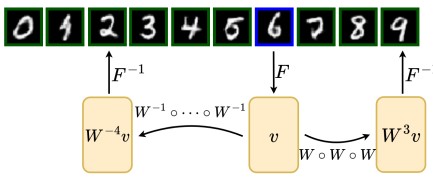

Figure 3: **MNIST Example:** Modelling the relationships between the digits as linear mappings in the target Category.

the latent code for 6, hops backwards 4 times and generates an image of 2. All images except "6" shown in Fig. 3 were generated in this way. It is worth making a brief comment on the generality here. If we had $k$ different morphisms, say, "$+1$" as well as rotation, scaling and shear, we could apply them in sequence and generate the corresponding images, even if such training data were not shown to the model (more experiments can be found in §A). This feature will allow dealing with multiple covariates easily in our experiments. A similar solution has been provided in Keurti et al. (2023), although their experiments were restricted to simple datasets and dealt only with 2D object dispositions. Here, we already showed how to learn something more complicated (addition) and, in the next section, we will show how to adapt this idea to complicated covariates.

## 4 A CATEGORY THEORY INSPIRED FORMULATION FOR DATA POOLING

**Overview.** The preceding section provides us all the necessary modules. Our task involves using brain MR images to predict diseased or healthy controls status. Our covariates will include secondary (non-imaging) data pertaining to the participants, including scanner type, site, age, sex, and genotype status. Some of these are ordinal/continuous such as genotype (APOE; three risk types) and age, whereas others are categorical such as scanner type, site and sex. For the categorical covariates, our formulation will seek to enforce *invariance* on the learned representations, i.e., asking that the latent representations be devoid of information pertinent to nuisance variables like scanner and site. It turns out that sex is associated with Alzheimer's disease (AD) because two-thirds of those diagnosed are women (Mielke, 2018). But if our goal is to understand which image features are relevant for the disease (and not simply to maximize accuracy), it makes sense to control for sex as a nuisance variable and add it separately at the last layer, if needed. For ordinal/continuous covariates, we see both shifts and imbalances across sites or scanner (different sites/scanners may not cover exactly the same age range of participants or include the same number of individuals for each genotype value). Equivariance will allow adjusting for these shifts, i.e., a coordinate system where the latent representations are "aligned" (modulo their covariate induced morphism). The diagram in Fig. 4 will setup our overall formulation, and we use the example below to describe its operation.

**Example 9** *Assume we want to control only for age when learning the latent representations of the images for objects/participants $S_1$ and $S_2$. $S_1, S_2$ are identical for all covariates but have different ages. Their scans are also different. If $S_2$ is $x$ years older than $S_1$, in $\mathcal{S}$, we have $S_2 \cong f_x(S_1)$ (i.e., equal up to isomorphism. In practice, isomorphism can be as strict as $\ell_2$-norm, cosine similarity as in CLIP-based models (Radford et al., 2021), or even distribution-based measures such as MMD). For someone, $2x$ years older, similar to Fig. 3, we will compose $f_x$ twice. In $\mathcal{T}$, latent representations $T_1, T_2$ correspond to $F(S_1)$ and $F(S_2)$. We seek to learn a functor $F$ by learning*

*$g_x \equiv F(f_x)$ such that $T_2 \cong g_x(T_1)$. If $S_1$ and $S_2$ differed in two covariates, by amounts $x$ and $y$ resp., the morphism from $S_1$ to $S_2$ would involve composing $f_x$ and $h_y$ (if $h$ denoted the morphisms for the second covariate). The morphisms in the Category $\mathcal{T}$ would compose similarly.*

**Setting up a loss function.** We will use simple morphisms in $\mathcal{T}$ which correspond to linear transformations $W \in \mathbb{R}^{n \times n}$ (this is a design choice). The pair of Functors $(F, F^{-1})$ correspond to an autoencoder, while the pair $(F, C)$ corresponds to an appropriate classifier (or regressor). Simply walking the paths from Fig. 4 provides all necessary constraints which we write as distinct terms in our loss function,

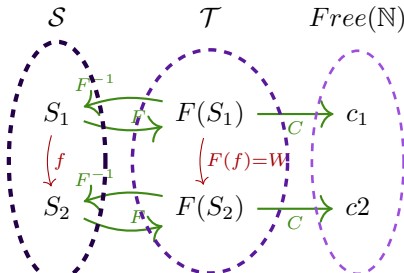

**(i)** $(F^{-1} \circ F)(x) = id_x$ gives the reconstruction loss:
$$\mathcal{L}_r = \sum_{s \in \mathcal{S}} \left\| s - (F^{-1} \circ F)(s) \right\|_2^2 \quad (5)$$

**(ii)** the prediction loss (given labels $\mathbf{y}$):
$$\mathcal{L}_p = \sum_{(i,s) \in \mathcal{S}} \underbrace{\mathcal{CE}\Big(\mathbf{y}_i, (C \circ F)(s)\Big)}_{\text{cross-entropy}} \quad (6)$$

**(iii)** Finally, preserving the morphisms, in its simplest form, is a structure-preserving loss,
$$\mathcal{L}_s = \sum_{s_1 \in S_1, s_2 \in S_2} \left\| W \cdot F(s_1) - F(s_2) \right\|_2^2 \quad (7)$$

Figure 4: A diagrammatic representation of equivariance with respect to a single covariate whose change corresponds to "$f$" in the original data space. Our goal is to preserve this structure in the latent space (Category $\mathcal{T}$) in which we model $F(f)$ as a linear transformation $W \in \mathbb{R}^{n \times n}$. In many practical cases the downstream goal is to classify the latent representations, which we formulate with the Functor $C : \mathcal{T} \to Free(\mathbb{N})$ (or $Free(\mathbb{R})$ for a regression task, etc.)

The training for multiple covariates (Alg. 1) is a direct generalization of the above formulation.

---

**Algorithm 1** Structure preserving training of Functors

**Input:** Parameters: $(\theta, \mathbf{W})$, Multipliers: $\lambda$
**Data:** input/output: $(S, \mathbf{y}) \in (\mathbb{R}^{m \times n}, \mathbb{N}^m)$, and covariates: $C \in \mathbb{N}^{m \times c}$

1   **for** *ep in epochs* **do**
2     $\mathcal{L}_r = \sum_{s \in S} \left\| s - (F_{\theta_1}^{-1} \circ F_{\theta_2})(s) \right\|_2^2, \mathcal{L}_p = \sum_{i=1}^m \mathcal{CE}(\mathbf{y}_i, (C_{\theta_3} \circ F_{\theta_1})(S_i)), \mathcal{L}_s = 0$
3     **for** $\mathbf{c} \in C$ **do**
4       **for** $(s_1, s_2), d$ *in pairs(S, $\mathbf{c}$)* **do**
5        $\mathcal{L}_s \mathrel{+}= \left\| W_{\mathbf{c}}^d \cdot F_{\theta_1}(s_1) - F_{\theta_1}(s_2) \right\|_2^2$
6     $\mathcal{L} = \lambda_1 \mathcal{L}_r + \lambda_2 \mathcal{L}_p + \lambda_3 \mathcal{L}_s$
7     $\text{step}\big(\mathcal{L}, (\theta_1, \theta_2, \theta_3, W_1, ..., W_c)\big)$     // update
8   **Function** `pairs`(S, $\mathbf{c}$)**:**
9     pairs = []
10    **for** *(i, j) in $m \times m$* **do**
11      **if** $S_i$ *is paired with* $S_j$ **then**
12       pairs.append$\big((S_i, S_j), \mathbf{c}_i - \mathbf{c}_j\big)$
13    **return** *pairs*
14

---

Enriching the latent space with structure provides some information about "hypotheticals". Given that we are unlikely to be provided different versions of a sample (one for each composition of the known Morphisms), a structure-preserving latent space gives us a way to:

**(i)** generate new data samples, by evaluating the expression $(F^{-1} \circ F(f) \circ F)(S)$.
**(ii)** answer questions, by evaluating the expression $(C \circ F(f) \circ F)(S)$.

**Role of Category Theory.** There is a clear advantage of expressing the problem in Category Theory. It enables a more concise and straightforward representation of the problem (sometimes, apparent in hindsight), as well as additional capabilities compared to earlier works (Lokhande et al., 2022).

## 5 EXPERIMENTAL EVALUATIONS

**Rationale.** Our motivation stems from pooling brain imaging datasets, particularly in scenarios where covariate distributions vary across sites. In the ADNI brain imaging study (Mueller et al.,

2005), while acquisition protocols are consistent among 50+ sites, demographic distributions differ. The diversity of scanners across sites, along with the impact of scanner upgrades on analysis tasks (Chen et al., 2020a; Ashford et al., 2021; Lee et al., 2019), further complicates the scenario. Scanner variations, even within the same study (e.g., ADNI-3), introduce controlled nuisances for standard analyses. Beyond ADNI, we also perform analysis on the ADCP dataset Lokhande et al. (2022). Our objective is to incorporate this functionality into representation learning for regression/classification.

This problem setting underscores limitations in existing methods. Recent VAE or GAN-based approaches (Ren et al., 2021; Hu et al., 2023; Moyer et al., 2020; Bashyam et al., 2022) target scanner invariance, often accommodating binary or multiple scanner variables. However, handling variations in other covariates (e.g., age, sex, APOE) remains challenging (Husain et al., 2021) (see §B.1).

**Baselines.** The following baselines address distributional differences in the data while maintaining associations with covariates: **1.** *Naive*: Pooling data without pre/post processing. **2.** *MMD* (Li et al., 2014): Minimizes Maximum Mean Discrepancy (MMD) for invariance but lacks equivariance. **3.** *CAI* (Xie et al., 2017): Achieves invariance with a discriminator but lacks equivariant mappings. **4.** *SS* (Zhou et al., 2017): Divides the population into subgroups (Subsampling-SS) and minimizes MMD for each subgroup. **5.** *RM* (Motiian et al., 2017): Matches similar samples from different scanners for invariance. **6.** *GE* (Group Equivariance) (Lokhande et al., 2022): Uses group theory to minimize MMD while seeking equivariance with respect to one covariate. GE achieves the best overall accuracy but has limitations: it uses expensive matrix exponentials and relies on a two-stage training, handling only two covariates.

**Evaluations.** As the classification task, we will predict Alzheimer's disease (AD)/Control normals (CN) labels using the invariant and/or equivariant representations. We will use the following metrics to systematically assess each algorithm. **(a)** Accuracy ($\mathcal{ACC}$): Test set accuracy of whether or not a participant has AD. Accuracy is reported in order to ensure that the model, despite its extra constraints, maintains all the required information that an input image carries. **(b)** $\mathcal{MMD}$: MMD is a measure of distributional differences, defined as the Euclidean distance of the kernel embeddings means (typically an RBF kernel). **(c)** $\mathcal{ADV}$: an alternative to measure invariance is by training a model to predict the nuisance covariate (e.g., "Scanner") using the latent representations. If we obtain latent representations devoid of this information, then the accuracy should be almost random. The above two metrics identify invariance with respect to scanners. To check equivariance to covariates, we use two measures: Minimum distance ($\mathcal{D}$) and Cosine similarity ($\mathcal{CS}$),

**(a)** Assume that a value of $c_1$ (for a covariate **c**) corresponds to an individual $s_1$, and Morphism $W \in \mathbb{R}^{n \times n}$ in the latent space gives a change in the covariate from $c_1$ to $c_2$. Then, the Minimum Distance (similar to covariate matching) identifies the closest individual $s$ such that $s_{\mathbf{c}} = c_2$:

$$\mathcal{D}(s_1; c_2) = \min_{s:s_{\mathbf{c}}=c_2} \frac{\left\| W \cdot F(s_1) - F(s) \right\|_2}{n} \tag{8}$$

**(b)** A low minimum distance by itself is insufficient. So, we also calculate the cosine similarity ($\mathcal{CS}$) between all the other covariates of $s_1$ and $s_2$, where $s_2 = \arg\min_{s:s_{\mathbf{c}}=c_2} \left\| W \cdot F(s_1) - F(s) \right\|_2$.

Low Minimum Distance ($\mathcal{D}$) and high Cosine Similarity ($\mathcal{CS}$) is desirable.

**Setting.** We model the Functor $F$ using a modified ResNet (He et al., 2016), and the Functor $C$ using a Fully-Connected Neural network. For our experiments, using linear mappings $W \in \mathbb{R}^{n \times n}$ for the Morphisms in the target Category $\mathcal{S}$ (i.e., latent space) was sufficient but this can be easily upgraded. All the experiments are a result of 5-fold cross validation procedure. In the following sub-sections, we summarize our experimental findings. We gradually increase the complexity of the experiments (number of constraints that we simultaneously optimize).

**Can Category theory constraints help impose invariance to scanner?** The following objective derived directly from the formulation is used,

$$\mathcal{L} = \underbrace{\mathcal{L}_p}_{\text{cross-entropy}} + \sum_{s_1, s_2 \in \mathcal{S}} \lambda \cdot \underbrace{\left\| F(s_1) - F(s_2) \right\|}_{\forall\, s_1, s_2 \text{ from different Scanners}} \tag{9}$$

Our results (Table 2) show that we obtain invariant representations without degrading the model's accuracy, consistent with the literature on invariant representation learning (Moyer et al., 2018). MMD decreases by more than $80\%$ compared to the best baseline, while accuracy is $1.3\%$ higher than the naive model, suggesting stronger generalization to new samples (more experiments on tabular data can be found in §B.2, and ablation studies on $\lambda$ and latent space size in §B.3).

| | ADNI | | | ADCP | | |
|---|---|---|---|---|---|---|
| | $\mathcal{ACC} \uparrow$ | $\mathcal{MMD}(\times 10^2) \downarrow$ | $\mathcal{ADV} \downarrow$ | $\mathcal{ACC} \uparrow$ | $\mathcal{MMD}(\times 10^2) \downarrow$ | $\mathcal{ADV} \downarrow$ |
| Random | 64 | - | 49 | 74 | - | 42 |
| *Naive* | $80_{(2.6)}$ | $27_{(1.6)}$ | $59_{(2.9)}$ | $83_{(4.4)}$ | $90_{(08.7)}$ | $49_{(08.4)}$ |
| *MMD*(Li et al., 2014) | $80_{(2.6)}$ | $27_{(1.8)}$ | $59_{(3.3)}$ | $84_{(6.5)}$ | $86_{(11.0)}$ | $49_{(11.9)}$ |
| *CAI*(Xie et al., 2017) | $74_{(3.6)}$ | $27_{(1.5)}$ | $61_{(2.1)}$ | $82_{(5.1)}$ | $85_{(12.3)}$ | $56_{(06.9)}$ |
| *SS*(Zhou et al., 2017) | $81_{(3.7)}$ | $26_{(1.6)}$ | $57_{(2.1)}$ | $82_{(3.5)}$ | $88_{(14.6)}$ | $51_{(06.7)}$ |
| *RM*(Motiian et al., 2017) | $78_{(3.8)}$ | $22_{(0.6)}$ | $52_{(5.4)}$ | $84_{(5.3)}$ | $77_{(13.8)}$ | $\mathbf{40}_{(04.7)}$ |
| *GE*(Lokhande et al., 2022) | $77_{(4.8)}$ | $16_{(7.2)}$ | $\mathbf{50}_{(4.2)}$ | $81_{(1.8)}$ | $70_{(22.3)}$ | $49_{(07.3)}$ |
| **Ours** (inv) | $81_{(2.3)}$ | $\mathbf{02}_{(2.4)}$ | $52_{(2.0)}$ | $\mathbf{86}_{(8.0)}$ | $\mathbf{34}_{(01.2)}$ | $45_{(04.2)}$ |
| **Ours** (1 cov) | $\mathbf{82}_{(2.5)}$ | $\mathbf{11}_{(2.9)}$ | $53_{(1.2)}$ | $\mathbf{86}_{(8.0)}$ | $\mathbf{39}_{(03.6)}$ | $48_{(05.7)}$ |
| **Ours** (2 cov) | $\mathbf{82}_{(2.2)}$ | $\mathbf{10}_{(6.6)}$ | $51_{(3.0)}$ | $85_{(7.7)}$ | $\mathbf{40}_{(04.3)}$ | $44_{(08.1)}$ |
| **Ours** (5 cov) | $80_{(2.3)}$ | $\mathbf{11}_{(5.3)}$ | $52_{(1.7)}$ | — | — | — |

Table 2: **Quantitative Results on ADNI (Mueller et al., 2005) and ADCP from (Lokhande et al., 2022).** The mean accuracy ($\mathcal{ACC}$) and invariance as evaluated by $\mathcal{MMD}$ and $\mathcal{ADV}$ are shown. The standard deviations are in parenthesis. The baseline *inv* corresponds to only invariance to Scanner, *x cov* corresponds to equivariance with respect to $x$ covariates (along with invariance to Scanner). $\mathcal{MMD}$ measure is significantly reduced without any drop in accuracy. The $\mathcal{ADV}$ measure is close to the random baseline (desirable).

**Can a model be invariant to scanner and remain equivariant to covariates?** Besides invariance, we seek equivariance to specific covariates in some cases. Here, we examine the simplest case of equivariance to a single covariate. We test three different covariates: 1. Age 2. Sex 3. APOE A1. Since age is continuous, we discretize it into bins of 10 years (so we have enough samples in each bin). APOE A1 (and A2) are genotypes in ADNI that are associated with AD (Husain et al., 2021; Sienski et al., 2021). Using our formulation, we derive the following loss function,

$$\mathcal{L} = \underbrace{\mathcal{L}_p}_{\text{cross-entropy}} + \sum_{s_1, s_2 \in \mathcal{S}} \lambda \underbrace{\left\| W^{c_{s_1} - c_{s_2}} \cdot F(s_1) - F(s_2) \right\|}_{c_{s_1}, c_{s_2} \text{ represent the nuisance covariate}} \qquad (10)$$

Note that the invariance term is implicit in this case because for two individuals $s_1, s_2$ with $c_{s_1} = c_{s_2}$ we recover the invariance term from (13).

**Remark 10** *While this specific problem setting is not new (see (Motiian et al., 2017)), here we show that the simplicity of our method comes with certain **advantages**. First, we get improved results and further, our model has fewer parameters and smaller runtime complexity (Tables 2, 1).*

In Fig. 5, we examine the performance of the naive model where we infer $W$ post-training, GE (Lokhande et al., 2022) which defines a different linear transformation $W$ for each age difference, and our model (Ours). In all three models, we model the age increase (e.g., 60 to 65) by applying a linear transformation in the latent space. Then, we find the closest point for that age (e.g., 65) in the latent space w.r.t. $\ell_2$ norm ($\mathcal{D}$) and cosine similarity ($\mathcal{CS}$). The transformed vector should be in the neighborhood of latent vectors with that age

| | Parameters | | | Runtime |
|---|---|---|---|---|
| | Encoder | Classifier | Morphisms | |
| GE | $\mathcal{O}(\mathcal{E})$ | $\mathcal{O}(\mathcal{C})$ | $\mathcal{O}(\mathbf{c} \cdot \mathcal{P})$ | $\mathcal{O}(\mathbf{c} \cdot \mathcal{R})$ |
| **Ours** | $\mathcal{O}(\mathcal{E})$ | $\mathcal{O}(\mathcal{C})$ | $\mathcal{O}(\mathbf{c})$ | $\mathcal{O}(\mathcal{R})$ |

Table 1: **Parameters / Runtime comparison with respect to the number of covariates. c** is the number of covariates and $\mathcal{P}$ the number of unique pairs for each covariate (exponential growth). $\mathcal{E}/\mathcal{C}$ is the number of parameters in the Encoder and Classifier respectively (independent of **c**). $\mathcal{R}$ stands for a single training runtime. Our method's complexity is better both in parameters and runtime.

(e.g., 65), and so $\mathcal{D}$ should be small and $\mathcal{CS}$ should be high. The results show that such a morphism is not accurate enough in the naive or even the GE algorithm. The naive model tends to "spread" the latent vectors without preserving structure, so it cannot capture equivariance in the latent space (clear when we examine $\mathcal{D}$), while GE achieves equivariance but our results suggest improvements.

**Assessing hypothetical samples.** Enriching the latent space with structure provides information about questions such as *What would be the change in AD if APOE A1 had a different value?* Recall that APOE A1 is a ordinal covariate with 3 values ($\{2, 3, 4\}$) and associated with AD (Husain et al., 2021; Sienski et al., 2021). For each sub-group (individuals with a specific value for APOE A1), we increase/decrease its value using morphisms and then classify the new vector. An increase in APOE A1 results in a higher probability of AD and vice-versa (Fig. 6), as expected (Husain et al., 2021).

# 6  RELATED WORK

**Data pooling, Fairness, disentanglement, and Invariant Representation Learning.** General approaches for analyzing data from different sites involve meta analysis (Thomp-

son et al., 2014; Rücker et al., 2021). When data transfer is feasible, pooling can be approached using the Johnson-Neyman technique (e.g., when ANCOVA is inapplicable). Some tools from statistical genomics have been deployed for brain image data pooling (Johnson et al., 2007), also see (Garcia-Dias et al., 2020). For deep models, the link between invariant representation learning/fairness and data pooling was seen in (Moyer et al., 2018; Jaiswal et al., 2019; Banijamali et al., 2017), and has been the defacto approach (Moyer et al., 2020; Lokhande et al., 2022; Peng et al., 2017; Liu et al., 2022; Pham et al., 2023) to disentangle the influence of nuisance variables (but provably doing so is difficult (Locatello et al., 2022)).

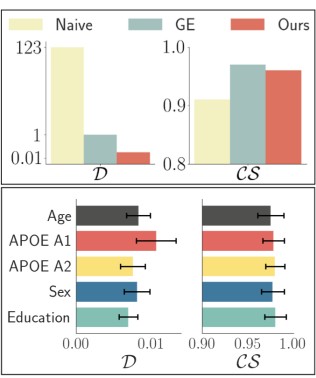

**Category theory in machine learning.** The applications of Category theory in applied disciplines are somewhat limited. But more recently, these ideas have been successfully applied to other fields (e.g. (Fong & Spivak, 2019)). In machine learning specifically, (Fong et al., 2019; Cruttwell et al., 2022; Fong & Johnson, 2019; Barbiero et al., 2023) model the learning process of gradient-based learning algorithms using Lenses and Gavranović et al. (2024) uses Monads. Separately, (Wilson & Zanasi, 2020) used Category theory to offer a framework for training Boolean circuits. The results in (Gavranović , 2020) modeled CycleGAN using Functors and showed how the formulation can be used for inserting/deleting objects from an image. A summary of developments is in (Shiebler et al., 2021).

Figure 5: **(Left)** Minimum Distance ($\mathcal{D}$) and Cosine Similarity ($\mathcal{CS}$) in for age equivariance, compared with Naive and GE. **(Right)** Minimum Distance ($\mathcal{D}$) and Cosine Similarity ($\mathcal{CS}$) for 5-covariate equivariance. Results are consistently good.

## 7 CONCLUSIONS

Imposing structure on latent spaces learned by DNN models is being actively studied, for problems ranging from disentanglement to interpretability. Using a data pooling problem in brain imaging as a motivation, we discuss how category theory provides a precise framework for such tasks. Not only does such an approach significantly simplify recent works in the literature, but also offers a perspective unifying an array of standalone approaches/models. Due to its rather abstract nature, the application of category theory in vision/machine learning is

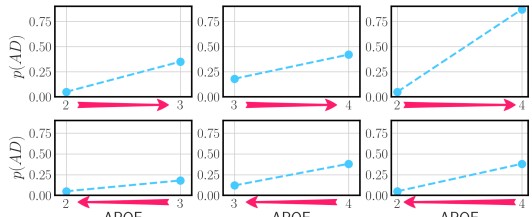

Figure 6: Increasing APOE A1 in the latent space using morphisms leads to a higher probability of AD and vice versa. Such manipulations in latent space are feasible through learned morphisms.

rather limited. Nonetheless, we believe that the models/experiments in this paper provide evidence that these ideas can inform other challenging problems in our field, including explainability, modularity and interpretability.

**Limitations**. We point out a few key caveats. Since the goal was to demonstrate the benefits of the formalism rather than maximize accuracy attained for each downstream task, the choice of most modules was kept quite simple (linear transformations, ResNets etc). Our construction offers a great deal of flexibility to upgrade these modules, as needed. However, due to the nature of our algorithm (i.e., imposing constraints on the latent space), we cannot directly benefit from contemporary architectures such as U-Net (Ronneberger et al., 2015; Zuo et al., 2021; Zhang et al., 2018) since their skip-connections provide shortcuts that will avoid the category-theory informed constraints. This means that, in contrast to some other works that harmonize two sets of images (mostly, with one categorical nuisance variable), the proposed method is best suited for obtaining a well-behaved latent space with multiple nuisance variables/covariate shifts, for a broad range of downstream tasks.

**Acknowledgments**. The authors thanks Veena Nair and Vivek Prabhakaran from UW Health for help with the ADCP dataset. Research was supported by NIH grants RF1AG059312 and RF1AG059869, as well as NSF award CCF 1918211 and the Bodosaki scholarship.

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

# A  A WARM-UP ANALYSIS

## A.1  MULTI-EQUIVARIANCE

In this section, we evaluate the performance of our approach when we require the latent space to be equivariant with respect to two different transformations. We consider a dataset of images and the goal is to create representations that are equivariant with respect to image rotations as well as image scaling.

We use the MNIST dataset LeCun et al. (1998) as a toy example to illustrate the key idea. We assume that the latent space is the Category with the following characteristics:

**(a)** Objects: vectors $\in \mathbb{R}^n$
**(b)** Morphisms: orthogonal linear transformations $W \in \mathbb{R}^{n \times n}$, $W^T W = I$ (the identity morhpism is the identity matrix $I$)

We learn an autoencoder (i.e., a pair of fully faithful Functors $F, F^{-1}$) and two matrices $W_r, W_s$ that represent the rotation and scale morhpism respectively in $\mathbb{R}^n$. In this experiment, we set $n = 128$. During training, we provide our model with an image from MNIST, along with a rotated version (of the same image) and separately, a scaled one. An important note is that **we do not provide any images that are both rotated and scaled during training.** We define

1. *Rotate* to be a counter-clockwise rotation by 5 degrees. This means that a rotation by e.g. 15 degrees corresponds to "*Rotate ∘ Rotate ∘ Rotate*" in the source Category of images and to $W_r^3$ in the latent space of $\mathbb{R}^n$
2. *Scale* the operation of "zoom-out" in an image. To achieve this behaviour, we first pad the input image (1 extra pixel on each side) and then we resize it back to the original shape. According to this definition, the inverse operation ($Scale^{-1}$) is defined as zooming into the image.

Our training objective consists of the following loss terms:

$$\mathcal{L} = \mathcal{L}_r + \lambda \mathcal{L}_s + \mu \Big( (W_r^T W - I)^2 + (W_s^T W - I)^2 \Big) \tag{11}$$

The first two terms are suggested directly by our framework while the last two constraints ensure that the latent space morphisms have the desired form and are case-specific.

Although during training we provide images that are only counter-clockwise rotated (up to 50 degrees) or scaled (up to 10 paddings), our model is able to map the images to the latent space and preserve the two morphisms, which leads to the following abilities:

**(a)** We can generate realistic images that are counter-clockwise rotated by more than 50 degrees by continuously applying the linear transformation $W_r$ in the latent space and mapping the new vector back to the original Category of images using the decoder ($F^{-1}$).

**(b)** We can generate realistic images that are scaled by more than 10 pads by continuously applying the linear transformation $W_s$ in the latent space and mapping the new vector back to the original Category of images using the decoder ($F^{-1}$).

**(c)** We can generate realistic images that are clockwise rotated by applying the inverse linear transformation $W_r^{-1}$ to the latent representation of an image and then map the new vector back to the original Category of images using the decoder ($F^{-1}$).

**(d)** We can generate realistic images that are zoomed-in by applying the inverse linear transformation $W_s^{-1}$ to the latent representation of an image and then mapping the new vector back to the original Category of images using the decoder ($F^{-1}$).

**(e)** Finally, we can generate realistic images that are both rotated and scaled by applying both linear transformations $W_r \circ W_s \ (= W_s \circ W_r)$ in the latent representation of an image and then mapping the new vector back to the original Category of images using the decoder ($F^{-1}$).

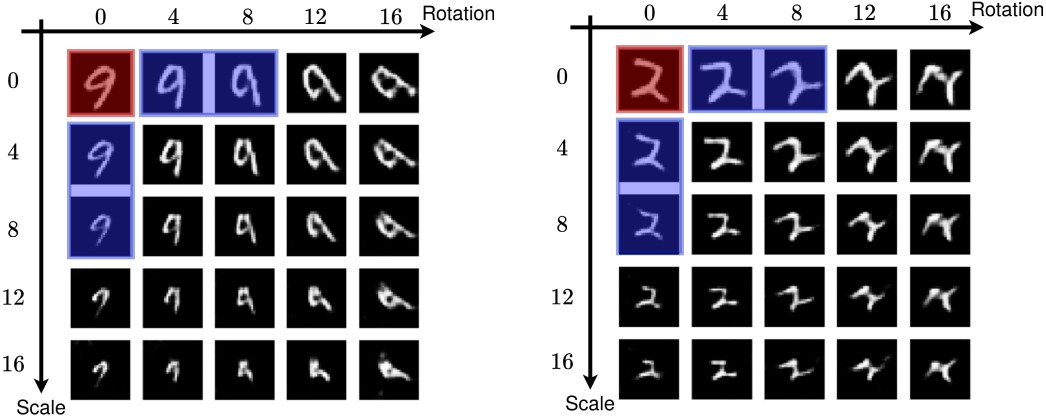

Figure 7: Composition of rotation and scaling. **The transformations were applied to the original image's latent vector** ($v$) **in the latent space** according to the composition rule $W_r^i W_s^j v$ where $i, j$ denote the degree of rotation and scaling respectively. In red we depict the original image, in blue we depict the only transformations that were "visible" during training.

### A.1.1 QUALITATIVE RESULTS

In Figure 7, we show the generated images when we apply both transformations to a latent vector. The 2 main takeaways from that figure are:

**(a)** Our model is able to generalize beyond the presented transformations as depicted by the row/column in which scale/rotation was $0$ respectively. We observe that even after applying $W_r^{20}, W_s^{20}$ (twice as much as the data presented during training) we get realistic results that follow the rotation and scaling rules.

**(b)** Our model is able to compose the transformation in the latent space and generate images that follow both transformations, although no composed images were presented during training.

In Fig. 8, the effect of the inverse linear transformation is shown. Although there were no such images during training, our model is able to understand the two transformations due to the categorical structure imposed on the latent space. As a result, we can generate clockwise rotated and zoomed-in images with no quality drop (zoom-in fails after a significant amount of constant applications due to the space constraints while rotation remains realistic even for an extreme number of linear transformations).

### A.1.2 QUANTITATIVE RESULTS

In Fig. 9 we quantify what we observed already. Rotation provides consistently good results independent of the direction, while inverse scaling (zoom-in) fails after some steps, due to the fact that the depicted digit is too big to fit in the image. On the contrary, we can continuously apply the scale operation (zoom-out) up to the point which the digit is no larger than a couple of pixels, with no decrease in the quality. Also, the results do not vary much as we increase the latent dimension from 32 to 256.

## A.2 EQUIVARIANCE TO ALGEBRAIC MORPHISMS

Here we show that the same formulation, without any change, is able to create a mapping to the latent space which is equiavariant to a much more abstract and complex morphism (algebraic addition). Specifically, we consider the MNIST dataset and our goal is to map the images to a latent space that preserves the relationships between the digits. We use the same Category as above for our latent space and we wish to enforce the following condition: if for two images, $I_1, I_2$ with labels $l_1, l_2$ the corresponding latent representations are $v_1, v_2$, then $v_2 \simeq W^{l_2 - l_1} v_1$.

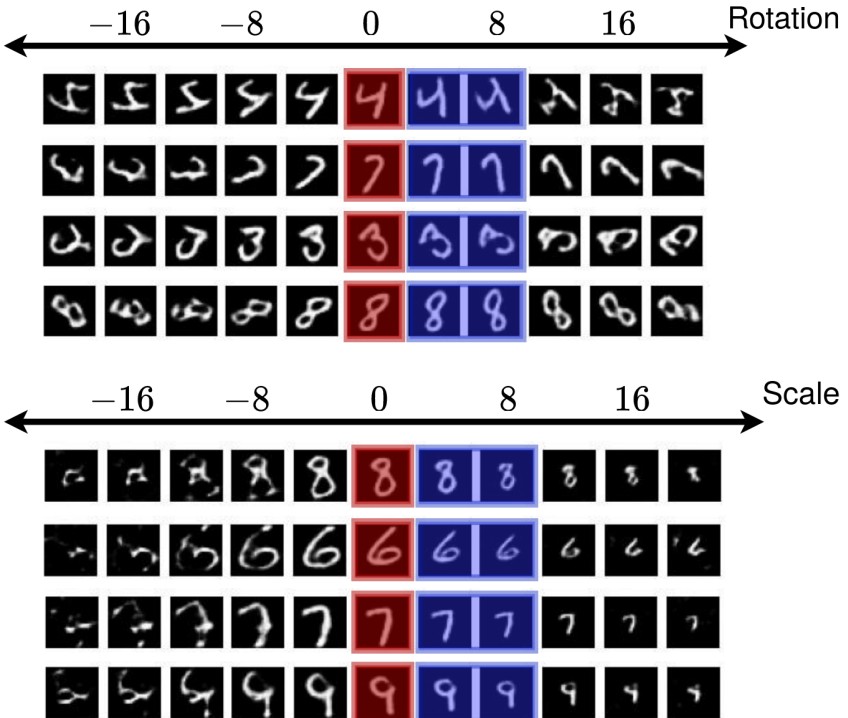

Figure 8: Rotation and Scaling in both directions. **The transformations were applied to the original image's latent vector** $(v)$ **in the latent space** according to the morphisms $W_r$, $W_r^{-1}$, $W_s$, $W_s^{-1}$. In red we depict the original image, in blue we depict the only transformations that were "visible" during training. Inverse scaling, as expected, fails after an adequate number of steps, since the digit can no longer fit in the image. On the contrary, rotation is more robust and performs well even for large rotations in both directions.

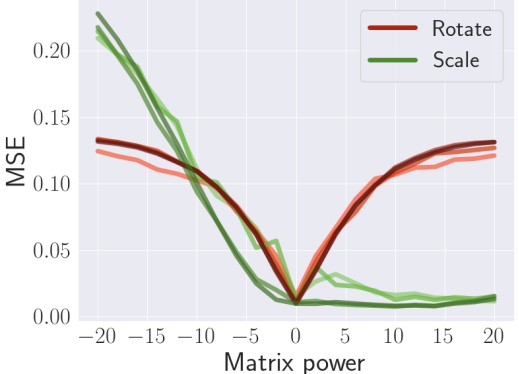

Figure 9: MSE error of the predicted image against the ground truth as we rotate it using $W_r, W_r^{-1}$ and scale it using $W_s, W_s^{-1}$. The line intensity represents the size of the latent space $(32, 64, 128, 256)$ and it shows that the results are accurate enough even for small latent spaces.

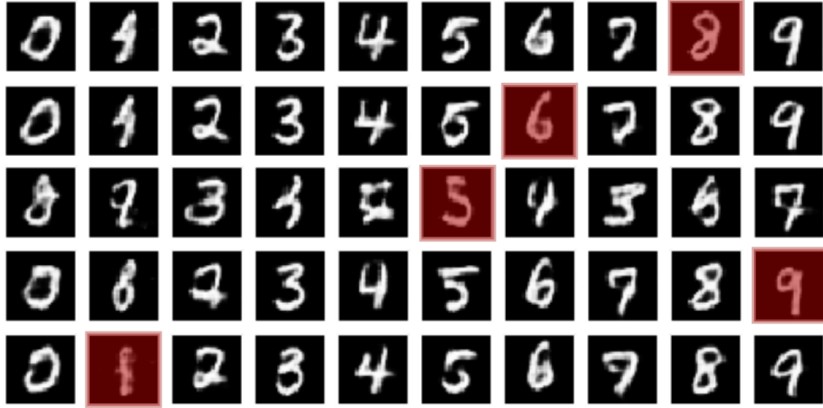

Figure 10: Images generated when we apply $W$ and $W^{-1}$ to the latent vector of the image in red. The images to the left of the red image were generated by using the linear transformation $W^{-1}$ on $v$, where $v$ is the latent vector of the original image (red), and the images to the right were generated by using the linear transformation $W$ on $v$. While there might exist some failed cases (e.g. row 3), in most cases we are able to generate realistic images for each digit.

We use the same setting as in the previous experiment (but here we learn a single matrix $W$). During training, we provide our model with pairs of data that corresponds to two images $(I_1, I_2)$ with $l_1 + 1 = l_2$. This means that, during training, we do not optimize for $W^{-1}$ simultaneously (just like in the previous experiment). In this experiment we set $n = 32$.

We use the same AutoEncoder architecture and the objective in this case is:

$$\mathcal{L} = \mathcal{L}_r + \lambda \mathcal{L}_s + + \mu\big((W^T W - I)^2\big) \tag{12}$$

which means that we use the exact same constraints that our formulation has suggested along with the constraint that $W$ is orthogonal.

### A.2.1 Qualitative results

Figure 10 shows some of the generated images we obtain when we apply the linear transformations $W, W^{-1}$ in the latent vector of an image. We can observe that, although we do not explicitly train our model for the inverse transformation $(W^{-1})$, it is able to generate realistic images due to the orthogonality constraint we impose in our model.

**Importance of orthogonality** Besides the structure preserving constraints we use another constraint in our models; the orthogonality constraint. In Fig. 11 we depict the results when such constraint is not enforced in the model. While still we can generate realistic images when we apply the transformation $W$ in the latent space, the results are less ideal when we apply the inverse transformation $W^{-1}$. Since we provide no constraints on $W$ during training, it ends up being a non-singular matrix. This means that we can only approximate the inverse matrix $W^{-1}$ and, as the results indicate, this estimation is not close to the ideal. In contrast, the orthogonality constraint forces the matrix to have a well-defined inverse $(W^{-1} = W^T)$.

**Smooth image interpolation** Since $W$ represents the abstract morphism *"add one"* we can assume that the matrix $W^a$ represents the abstract morphism *"add a"*, $\forall a \in \mathbb{R}$. This provides us with a simple way to interpolate between the digits and create smooth transitions.

In the supplementary material we have included .*gif* files that show a complete transition from $0$ to $9$ with a step of $0.2$. The results show that, although our goal was to preserve a specific relationship of the original data into the latent space, a by-product of our method is the ability of image interpolation and smooth transitions.

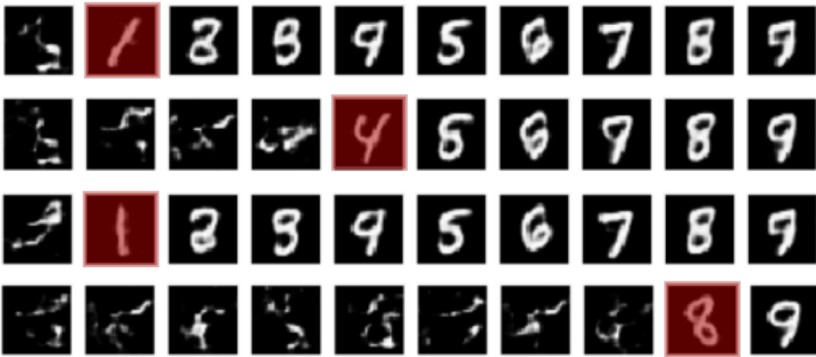

Figure 11: Without the orthogonality constraint, the inverse matrix may not exist and its approximation leads to generated images that do not resemble real digits (the application of the matrix $W$ though still leads to realistic images).

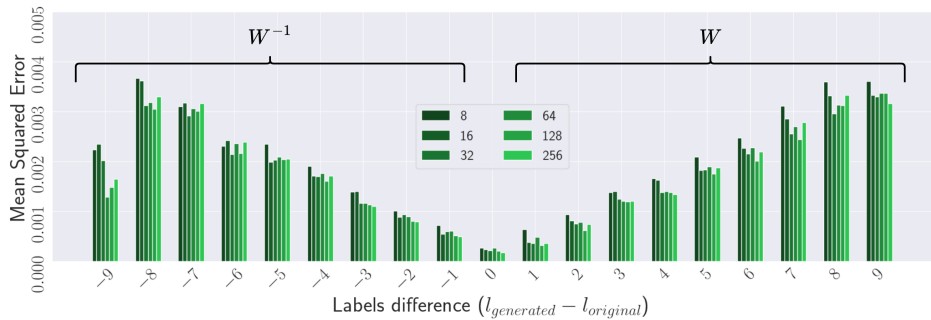

Figure 12: Mean Squared Error of the generated images for different sizes of the latent space ($n$). The application of the inverse has an only marginally higher MSE due to the orthogonality of the matrix $W$.

#### A.2.2 QUANTITATIVE RESULTS

Besides the qualitative results that indicate that our model successfully learned the relationship between the depicted digits, we can also quantify how accurate our model is. In Fig. 12 we show how the Mean Squared Error (MSE) changes when we continuously apply the linear transformations $W, W^{-1}$ to the latent vector and generate an image. In order to calculate it, we found the minimum MSE between the generated image and any image that depicts the specific digit. We see clearly that our model is able to perfectly generate an image of the new digit, no matter if it is smaller ($W^{-1}$ applied) or greater ($W$ applied), even after applying multiple times the linear transformations. Another important aspect is the robustness of our model to different values of $n$ (latent space dimension).

## B EXPERIMENTAL EVALUATIONS - IN DEPTH

### B.1 DATASET DETAILS

The ADNI dataset can be obtained from `https://adni.loni.usc.edu/`. The ADNI project was launched in 2004.

This specific dataset consists of 449 MRI scans, along with a dataset of covariates for each individual participant. Each MRI was obtained in one of three different scanners ({GE Medical Systems, Philips Medical Systems, SIEMENS}) and some of the covariates include {Age, Sex, Years of Education, APOE A1, APOE A2, Marital status, Race} along with a lot of cognitive battery scores such as {MMSE, RAVLT, Ecog}.

| | $\mathcal{ACC}\uparrow$ | $\mathcal{MMD}(\times 10^2)\downarrow$ | $\mathcal{ADV}\downarrow$ |
|---|---|---|---|
| *Naive* | $74_{(0.9)}$ | $7.7_{(0.8)}$ | $62_{(3.1)}$ |
| *MMD*Li et al. (2014) | $73_{(1.5)}$ | $1.5_{(0.3)}$ | $66_{(0.04)}$ |
| *CAI*Xie et al. (2017) | $76_{(1.3)}$ | $1.2_{(2.4)}$ | $65_{(0.01)}$ |
| *SS*Zhou et al. (2017) | $\mathbf{76}_{(0.9)}$ | $1.5_{(0.6)}$ | $70_{(6.9)}$ |
| *RM*Motiian et al. (2017) | $74_{(2.1)}$ | $7.5_{(0.9)}$ | $66_{(4.2)}$ |
| *GE*Lokhande et al. (2022) | $75_{(3.3)}$ | $2.7_{(0.6)}$ | $54_{(1.1)}$ |
| **Ours** (inv) | $74_{(0.6)}$ | $\mathbf{0.7}_{(0.1)}$ | $\mathbf{51}_{(2.4)}$ |
| **Ours** (1 cov) | $74_{(1.4)}$ | $1.3_{(0.9)}$ | $55_{(1.4)}$ |

Table 3: **Quantitative Results on German Hofmann (1994).** The mean accuracy ($\mathcal{ACC}$) and invariance as evaluated by $\mathcal{MMD}$ and $\mathcal{ADV}$ are shown. The standard deviations are in parenthesis. The baseline *inv* corresponds to only invariance to *foreigner*, 1 *cov* corresponds to equivariance with respect to the *age* covariates (plus invariance to *foreigner*). $\mathcal{MMD}$ and $\mathcal{ADV}$ are significantly reduced without any significant drop in accuracy.

In neuroimaging data, "APOE" typically refers to the apolipoprotein E gene. The APOE gene is involved in encoding a protein that plays a crucial role in the metabolism of lipids (fats) in the body, including cholesterol. This gene has different variants, or alleles, known as APOE $\epsilon 2$, APOE $\epsilon 3$, and APOE $\epsilon 4$. In the context of neuroimaging and neuroscience, the APOE gene has been of particular interest because one of its alleles, APOE $\epsilon 4$, is considered a significant genetic risk factor for Alzheimer's disease Husain et al. (2021); Sienski et al. (2021).

The second dataset (ADCP) was shared with us by the authors of Lokhande et al. (2022).

The data for ADCP was collected through an NIH-sponsored Alzheimer's Disease Connectome Project (ADCP) U01 AG051216. The study inclusion criteria for AD (Alzheimer's disease) / MCI (Mild Cognitive Impairment) patients consisted of age between 55-90 years, willing and able to undergo all procedures, retaining decisional capacity at the initial visit, and meet criteria for probable AD or MCI. The MRI images were acquired at three distinct sites.

Besides the medical image datasets, we evaluate our performance in two tabular datasets; the German(Hofmann, 1994) and the Adult Becker & Kohavi (1996). The German consists of $58$ features, with some of them being *foreigner* and *age*, and our goal is to predict the consumers' default loans. The Adult consists of $100$ with sensitive attributes such as *sex* and *age*. Our goal here is to predict whether someone's income is higher than \$50K.

## B.2   Results on tabular data

Tables 3, 4 contains all 3 metrics for German and Adult datasets, respectively. We observe that we have accuracy similar to the other methods, while we have a significantly lower value of $\mathcal{MMD}$ and $\mathcal{ADV}$, meaning that we obtain a less biased embedding space and, as a result, predictions.

In all of our experiments we used a latent space of size 32 and a 1-hidden-layer (of size 32) feedforward network to map the input features to $\mathbb{R}^{32}$.

## B.3   Effect of Lagrange multiplier $(\lambda)$ and latent space dimension $(n)$

Here, we evaluate the effect of the latent space dimension as well as the effect of the Lagrange multiplier $\lambda$. As a reminder, the objective was defined as:

$$\mathcal{L} = \underbrace{\mathcal{L}_p}_{\text{cross-entropy}} + \sum_{s_1,s_2\in\mathcal{S}} \underbrace{\lambda\cdot\left\|F(s_1)-F(s_2)\right\|}_{\forall\ s_1,s_2\ \text{from different Scanners}} \tag{13}$$

in the case of invariance, and as:

| | $\mathcal{ACC} \uparrow$ | $\mathcal{MMD}(\times 10^2) \downarrow$ | $\mathcal{ADV} \downarrow$ |
|---|---|---|---|
| *Naive* | $84_{(0.1)}$ | $9.8_{(0.3)}$ | $83_{(0.1)}$ |
| *MMD*Li et al. (2014) | $84_{(0.1)}$ | $3.1_{(0.3)}$ | $83_{(0.1)}$ |
| *CAI*Xie et al. (2017) | $84_{(0.04)}$ | $2.2_{(2.4)}$ | $81_{(0.7)}$ |
| *SS*Zhou et al. (2017) | $84_{(0.1)}$ | $1.5_{(0.2)}$ | $83_{(0.2)}$ |
| *RM*Motiian et al. (2017) | $84_{(0.3)}$ | $4.8_{(0.7)}$ | $82_{(0.4)}$ |
| *GE*Lokhande et al. (2022) | $83_{(0.1)}$ | $7.1_{(0.6)}$ | $75_{(1.4)}$ |
| **Ours** (inv) | $83_{(0.2)}$ | $\mathbf{0.6}_{(0.4)}$ | $\mathbf{74}_{(1.6)}$ |
| **Ours** (1 cov) | $84_{(0.1)}$ | $\mathbf{1.1}_{(0.9)}$ | $76_{(1.3)}$ |

Table 4: **Quantitative Results on Adult Becker & Kohavi (1996).** The mean accuracy ($\mathcal{ACC}$) and invariance as evaluated by $\mathcal{MMD}$ and $\mathcal{ADV}$ are shown. The standard deviations are in parenthesis. The baseline *inv* corresponds to only invariance to *gender*, 1 *cov* corresponds to equivariance with respect to the *age* covariates (plus invariance to *gender*). Similarly to German, $\mathcal{MMD}$ and $\mathcal{ADV}$ are significantly reduced without any significant drop in accuracy.

$$\mathcal{L} = \underbrace{\mathcal{L}_p}_{\text{cross-entropy}} + \sum_{s_1,s_2 \in \mathcal{S}} \lambda \underbrace{\left\| W^{c_{s_1} - c_{s_2}} \cdot F(s_1) - F(s_2) \right\|}_{c_{s_1}, c_{s_2} \text{ represent the nuisance covariate}} \tag{14}$$

in the case of equivariance.

Tables 6, 10 present the results for 1-covariate equivariance, for both datasets. We can observe that moderate values of $\lambda$ (i.e. $\simeq 0.01$) lead to a low MMD value (much lower than the existing methods in most of the cases) with no Accuracy drop. In fact, in most of the cases, the effect of invariance leads to better generalization capabilities (i.e. higher Accuracy).

Tables 7, 11 show the same results but in the case of 2-covariate equivariance. Although, in theory, such problem is harder to solve, this does not translate to lower values of MMD and Accuracy. In fact, MMD is slightly lower in this case, since we enforce invariance to each subgroup separately (i.e. a less "aggressive" form of invariance).

Finally, in Table 8 and Fig. 13 we present the results for 5-covariate equivariance, for the ADNI dataset. The results show, undoubtedly, that we have presented a general method which has no problem in handling multiple nuisance covariates, in contrast to existing approaches that either can not handle such scenarios, or exhibit a high performance drop.

| | Only invariance | | | | | |
|---|---|---|---|---|---|---|
| | | $\mathcal{ACC}$ | | | $\mathcal{MMD}$ | |
| $\lambda$ | 16 | 32 | 64 | 16 | 32 | 64 |
| 0.001 | $81_{(2.3)}$ | $81_{(1.3)}$ | $84_{(3.9)}$ | $18_{(4.4)}$ | $21_{(0.6)}$ | $24_{(0.8)}$ |
| 0.01 | $81_{(2.3)}$ | $79_{(0.1)}$ | $81_{(1.3)}$ | $06_{(2.0)}$ | $17_{(5.0)}$ | $16_{(4.8)}$ |
| 0.1 | $81_{(4.0)}$ | $78_{(3.8)}$ | $83_{(0.6)}$ | $02_{(2.4)}$ | $07_{(3.3)}$ | $13_{(8.4)}$ |

Table 5: ADNI (Mueller et al., 2005): Effect of the Lagrange multiplier $\lambda$ and size of the latent space $n$ to Accuracy and Maximum Mean Discrepancy (MMD), where we enforce only invariance (with respect to the site) to the latent space. Higher values of $\lambda$ lead to lower MMD (more invariant representations). In all the combinations of $(\lambda, n)$ Accuracy remains high (in most of the cases higher than the baselines) while MMD is significantly lower than the baselines in most cases.

|  | $\lambda$ | Age | | | Sex | | | APOE A1 | | |
|---|---|---|---|---|---|---|---|---|---|---|
|  |  | 16 | 32 | 64 | 16 | 32 | 64 | 16 | 32 | 64 |
| ACC | 0.001 | $79_{(3.3)}$ | $80_{(5.7)}$ | $79_{(4.0)}$ | $80_{(3.3)}$ | $81_{(3.0)}$ | $80_{(3.4)}$ | $79_{(3.9)}$ | $80_{(1.7)}$ | $79_{(3.2)}$ |
| ACC | 0.01 | $82_{(2.5)}$ | $80_{(3.2)}$ | $81_{(3.4)}$ | $80_{(4.5)}$ | $80_{(2.2)}$ | $81_{(1.3)}$ | $82_{(1.7)}$ | $80_{(3.4)}$ | $81_{(5.0)}$ |
| ACC | 0.1 | $79_{(1.3)}$ | $79_{(3.2)}$ | $74_{(2.9)}$ | $78_{(3.9)}$ | $77_{(1.7)}$ | $77_{(1.7)}$ | $78_{(4.2)}$ | $77_{(1.1)}$ | $74_{(1.1)}$ |
| MMD | 0.001 | $27_{(4.4)}$ | $27_{(0.6)}$ | $27_{(0.8)}$ | $27_{(1.2)}$ | $26_{(1.0)}$ | $27_{(3.6)}$ | $28_{(0.7)}$ | $27_{(0.7)}$ | $27_{(0.1)}$ |
| MMD | 0.01 | $11_{(2.9)}$ | $15_{(6.5)}$ | $19_{(3.6)}$ | $17_{(6.3)}$ | $13_{(2.9)}$ | $20_{(6.1)}$ | $11_{(4.7)}$ | $15_{(2.8)}$ | $22_{(4.0)}$ |
| MMD | 0.1 | $04_{(0.7)}$ | $09_{(8.1)}$ | $16_{(2.5)}$ | $03_{(0.4)}$ | $07_{(6.0)}$ | $15_{(5.1)}$ | $04_{(0.6)}$ | $11_{(4.4)}$ | $14_{(1.4)}$ |

Table 6: ADNI (Mueller et al., 2005): Effect of the Lagrange multiplier $\lambda$ and size of the latent space $n$ to Accuracy and to Maximum Mean Discrepancy (MMD) in the case of a single covariate equivariance. No Accuracy drop is observed, while MMD is significantly lower than the baselines, in most of the experiments.

|  | $\lambda$ | Age-APOE A1 | | | Age-Sex | | | APOE A1-Sex | | |
|---|---|---|---|---|---|---|---|---|---|---|
|  |  | 16 | 32 | 64 | 16 | 32 | 64 | 16 | 32 | 64 |
| ACC | 0.001 | $80_{(1.1)}$ | $79_{(3.2)}$ | $80_{(3.4)}$ | $82_{(4.2)}$ | $82_{(2.3)}$ | $82_{(2.2)}$ | $83_{(1.3)}$ | $81_{(2.2)}$ | $80_{(2.8)}$ |
| ACC | 0.01 | $80_{(1.3)}$ | $81_{(1.7)}$ | $82_{(1.7)}$ | $80_{(1.1)}$ | $81_{(3.6)}$ | $82_{(3.0)}$ | $82_{(2.2)}$ | $79_{(3.0)}$ | $80_{(3.6)}$ |
| ACC | 0.1 | $78_{(2.2)}$ | $77_{(1.9)}$ | $75_{(2.8)}$ | $77_{(2.8)}$ | $77_{(0.6)}$ | $77_{(2.3)}$ | $77_{(4.2)}$ | $79_{(2.3)}$ | $76_{(0.6)}$ |
| MMD | 0.001 | $24_{(2.8)}$ | $26_{(0.4)}$ | $27_{(0.1)}$ | $22_{(1.1)}$ | $27_{(2.9)}$ | $27_{(0.4)}$ | $25_{(3.7)}$ | $26_{(8.0)}$ | $26_{(4.0)}$ |
| MMD | 0.01 | $07_{(1.2)}$ | $14_{(7.0)}$ | $18_{(5.4)}$ | $09_{(0.4)}$ | $17_{(3.9)}$ | $16_{(4.7)}$ | $10_{(6.6)}$ | $16_{(5.7)}$ | $15_{(6.0)}$ |
| MMD | 0.1 | $04_{(4.7)}$ | $07_{(0.9)}$ | $10_{(7.0)}$ | $2_{(0.7)}$ | $04_{(2.2)}$ | $08_{(3.4)}$ | $03_{(2.6)}$ | $09_{(9.0)}$ | $09_{(4.0)}$ |

Table 7: ADNI (Mueller et al., 2005): Effect of the Lagrange multiplier $\lambda$ and size of the latent space $n$ to Accuracy and to Maximum Mean Discrepancy (MMD) in the case of a two-covariate equivariance.

| | Age-Sex-APOE A1-APOE A2-Education | | | | | |
|---|---|---|---|---|---|---|
| | $\mathcal{ACC}$ | | | $\mathcal{MMD}$ | | |
| $\lambda$ | 16 | 32 | 64 | 16 | 32 | 64 |
| 0.001 | $82_{(4.2)}$ | $81_{(1.7)}$ | $82_{(1.1)}$ | $11_{(5.3)}$ | $21_{(2.5)}$ | $25_{(3.4)}$ |
| 0.01 | $80_{(2.3)}$ | $78_{(2.3)}$ | $77_{(1.7)}$ | $05_{(5.3)}$ | $09_{(8.4)}$ | $14_{(3.7)}$ |
| 0.1 | $78_{(1.9)}$ | $76_{(2.2)}$ | $78_{(0.6)}$ | $01_{(0.2)}$ | $03_{(0.4)}$ | $07_{(5.6)}$ |

Table 8: ADNI (Mueller et al., 2005): Effect of the Lagrange multiplier $\lambda$ and size of the latent space $n$ to Accuracy and Maximum Mean Discrepancy (MMD), in the case of equivariance with respect to five variables.

| | Only invariance | | | | | |
|---|---|---|---|---|---|---|
| | $\mathcal{ACC}$ | | | $\mathcal{MMD}$ | | |
| $\lambda$ | 16 | 32 | 64 | 16 | 32 | 64 |
| 0.001 | $82_{(2.2)}$ | $82_{(2.2)}$ | $83_{(4.4)}$ | $48_{(5.9)}$ | $53_{(4.8)}$ | $60_{(7.0)}$ |
| 0.01 | $83_{(2.2)}$ | $86_{(8.0)}$ | $885_{(3.9)}$ | $35_{(5.1)}$ | $34_{(1.2)}$ | $54_{(4.1)}$ |
| 0.1 | $81_{(3.8)}$ | $82_{(5.9)}$ | $85_{(7.7)}$ | $15_{(2.5)}$ | $25_{(2.2)}$ | $28_{(2.0)}$ |

Table 9: ADCP: Effect of the Lagrange multiplier $\lambda$ and size of the latent space $n$ to Accuracy and Maximum Mean Discrepancy (MMD), where we enforce only invariance (with respect to the site) to the latent space.

|       | $\lambda$ | Age 16 | 32 | 64 | Sex 16 | 32 | 64 |
|-------|-----------|--------|-----|-----|--------|-----|-----|
| *ACC* | 0.001 | $85_{(3.8)}$ | $83_{(2.2)}$ | $86_{(5.9)}$ | $83_{(4.4)}$ | $83_{(4.4)}$ | $83_{(5.8)}$ |
|       | 0.01  | $79_{(2.2)}$ | $82_{(8.0)}$ | $86_{(5.9)}$ | $82_{(4.4)}$ | $81_{(6.7)}$ | $85_{(3.8)}$ |
|       | 0.1   | $77_{(3.9)}$ | $79_{(4.4)}$ | $83_{(4.4)}$ | $81_{(0.2)}$ | $78_{(8.0)}$ | $83_{(4.4)}$ |
| *MMD* | 0.001 | $37_{(4.0)}$ | $51_{(5.6)}$ | $57_{(5.3)}$ | $37_{(4.8)}$ | $52_{(8.4)}$ | $55_{(3.0)}$ |
|       | 0.01  | $30_{(4.2)}$ | $35_{(3.3)}$ | $39_{(3.6)}$ | $30_{(5.0)}$ | $31_{(4.2)}$ | $34_{(5.8)}$ |
|       | 0.1   | $27_{(3.2)}$ | $27_{(5.8)}$ | $23_{(5.4)}$ | $23_{(5.3)}$ | $21_{(7.0)}$ | $46_{(4.2)}$ |

Table 10: ADCP: Effect of the Lagrange multiplier $\lambda$ and size of the latent space $n$ to Accuracy and to Maximum Mean Discrepancy (MMD) in the case of a single covariate equivariance.

|           | Age-Sex | | | | | |
|-----------|---------|--------|-----|--------|-----|-----|
|           | $\overline{\mathcal{ACC}}$ | | | $\overline{\mathcal{MMD}}$ | | |
| $\lambda$ | 16 | 32 | 64 | 16 | 32 | 64 |
| 0.001 | $83_{(5.8)}$ | $82_{(4.4)}$ | $82_{(9.6)}$ | $37_{(2.1)}$ | $42_{(9.4)}$ | $49_{(4.1)}$ |
| 0.01  | $85_{(7.7)}$ | $82_{(9.5)}$ | $85_{(3.8)}$ | $29_{(3.3)}$ | $32_{(1.2)}$ | $35_{(6.1)}$ |
| 0.1   | $79_{(2.2)}$ | $79_{(8.7)}$ | $83_{(4.4)}$ | $24_{(4.6)}$ | $27_{(4.8)}$ | $22_{(2.8)}$ |

Table 11: ADCP: Effect of the Lagrange multipler $\lambda$ and size of the latent space $n$ to Accuracy and Maximum Mean Discrepancy (MMD), in the case of two-covariates equivariance.

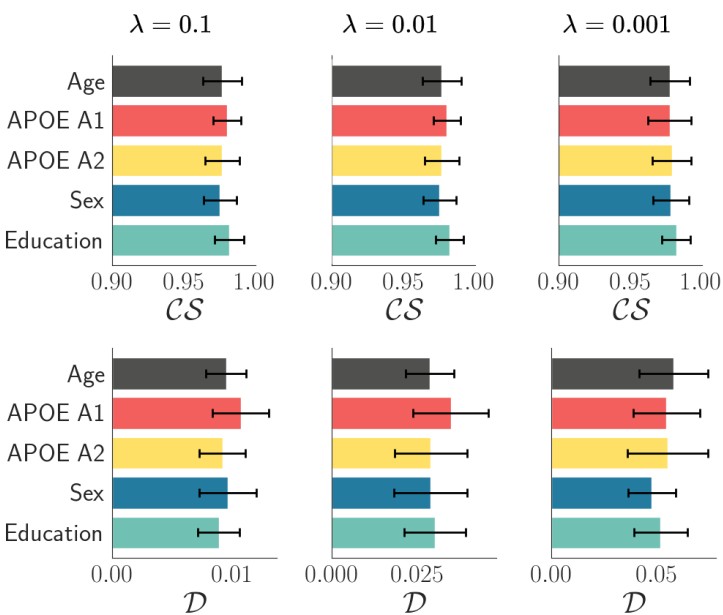

Figure 13: Effect of the Lagrange multipler $\lambda$ on the equivariance metrics $\mathcal{CS}$ and $\mathcal{D}$. As expected, $\lambda$ and $\mathcal{D}$ negatively correlated, while $\mathcal{CS}$ remains high for different values of $\lambda$.

