# OpenReview forum: "Pooling Image Datasets with Multiple Covariate Shift and Imbalance"
_ICLR.cc/2024/Conference — ICLR 2024 poster_

### Official Review · Reviewer_zBw9 · 2023-10-22

**Soundness:** 3 good
**Presentation:** 3 good
**Contribution:** 2 fair
**Rating:** 5
**Confidence:** 1

**Summary:**

This paper aims to provide a general harmonization tool that can handle multi-equivariance and multi-invariance with respect to the images' covariates.

**Strengths:**

The problem to be solved in the paper is more interesting.

**Weaknesses:**

I don't see any obvious Weaknesses

**Questions:**

None

---

> ### Author Response · Authors · 2023-11-22
>
> Dear Reviewer, please let us know if there are any questions we can answer. Thank you.

---

### Official Review · Reviewer_Fvtj · 2023-10-31

**Soundness:** 3 good
**Presentation:** 3 good
**Contribution:** 2 fair
**Rating:** 6
**Confidence:** 3

**Summary:**

In this study, the authors introduced the category theory to reinterpret the problem of medical data pooling.  The study firstly elaborated on the MNIST toy dataset for explaining the morphism composition imposed on the latent space, then conducted experiments on ADNI and ADCP datasets by testing on the scanner parameter. Compared to the closely-related GE method, the proposed approach achieved both low time complexity and better performance reflected on ACC and MMD measurements.

**Strengths:**

1. The introduction of category theory to the medical data pooling is novel and interesting.

2. Despite the complicated topic, the paper is well-written and well-structured, therefore quite easy to understand.

3. Both the runtime and quantitative results are improved over the GE method, which are quite impressive.

**Weaknesses:**

1. Though it is novel and interesting to introduce category theory to this problem, I am not sure if it is really necessary to do so. As far as I understand and please correct me if I am wrong, there are no additional theoretical novelties other than quoting existing definitions to the paper.

2. In terms of the scanner parameter discussed in the experiments, it is also very common to apply different kinds of normalization tricks to resolve the issue. Unfortunately, I didn't see relevant discussion or experiments presented in the paper.

**Questions:**

Please see the weakness section.

---

> ### Author Response · Authors · 2023-11-22
>
> Dear Reviewer, thank you for the detailed review. We answer all questions below.
>
> **Though it is novel and interesting to introduce category theory to this problem, I am not sure if it is really necessary to do so. As far as I understand and please correct me if I am wrong, there are no additional theoretical novelties other than quoting existing definitions to the paper.**
>
> Yes, this paper does not derive new results in Category Theory. Instead, as noted in the other reviews and this one, the goal is to deploy ideas from Category Theory to an interesting scientific problem, and show that various models already popular in the community emerge naturally with this framing. Given the limited practical use of category theory for any problem in machine learning, the formalization in this language is a key message of the paper. To make the paper self-contained and easy to read, the definitions were included because they make the discussion precise, and it is then easy to check that the formulation follows from those definitions.
>
> **In terms of the scanner parameter discussed in the experiments, it is also very common to apply different kinds of normalization tricks to resolve the issue. Unfortunately, I didn’t see relevant discussion or experiments in the paper.**
>
> Yes, this is correct. If a scanner is the only variable we must control for, some scale/location normalization as a pre-processing step may be a good first step to adjust global intensity changes. We included these works in the revised version of our discussion. The already discussed solutions describe how mutual information and discrepancy-based losses are more effective and improve performance. It is not clear how to pool while handling the multiple covariate case with a single normalization.
>
> Please let us know if we can answer any other questions.

---

### Official Review · Reviewer_SjKQ · 2023-10-31

**Soundness:** 4 excellent
**Presentation:** 3 good
**Contribution:** 4 excellent
**Rating:** 6
**Confidence:** 3

**Summary:**

The paper explores an interesting direction for giving a defined structure to the latent space by wrapping an understanding around it derived from category theory. The motivation is to find commonality among datasets in terms of the latent space and aid in data pooling / data harmonisation, something that is needed when data collection is expensive and data from multiple sources need to be combined.  The hope is that the theory is able to define structure and invariances/equivariances between them,, and invariance to shifts in feature distributions for clinical and demographic features when dealing with image data would be easily drawn. For quantifying  the shift, minimum distance and cosine sim are employed. The paper also describes a reinterpretaion of two differentiable models from adversarial and self-supervised learning families, and walks through a demonstration of latent space operation sequences on MNIST before developing notions of losses and evaluations on data. It reads almost like a position paper with few evaluations.

**Strengths:**

Even though elements of the ideas exist, as elaborated in literature, this is perhaps a first formulation of latent space equivalences in terms of category theory. The formulation expressed, developed and proved mathematically appears sound on a few reads of the text.

**Weaknesses:**

Both classes of functor mappings are linear or affine, given properties of identity, distributivity and commutativity. The development of these ideas for supra-linear mappings may present an uncertainty at the fundamental that is unknown at the moment.

**Questions:**

While converting to measurable spaces, does the scale of the Borel interval have an effect? That's something that may merit digging into.

While evaluating, accuracy seems just an indicator and not really a measure of distributional discrepancy like mmd and adv. Do I understand it right?

---

> ### Author Response · Authors · 2023-11-22
>
> Dear Reviewer,
>
> Thank you for the comments and questions. We provide detailed answers below,
>
>
> **Both classes of Functor mappings are linear or affine, given properties of identity, distributivity and commutativity.**
>
> We have indeed experimented with richer mappings, both for the Functor as well as for the morphisms. They work well. If recommended, we are happy to include the results in the Appendix. In the paper, linear/affine was chosen only to avoid detracting from the formulation, i.e., which benefits (say, in Fig. 3) were intrinsic to the formulation presented and is not attributable to the richness of modules being used for the Functor/Morphism classes.
>
> We should note the axioms/rules that we can read off from the diagram in Fig. 4 are otherwise not restrictive and can also be implemented via MLPs (or convolutional layers), e.g., as is already common in CycleGAN or SimCLR.
>
> **While converting to measurable spaces, does the scale of the Borel interval have an effect? That’s something that may merit digging into.**
>
> While some of our experiments report distances, our current construction does not construct a normalized measure on the latent space. We agree that doing so can allow saying more about the latent space structure learned by the category theoretic construction, similar to groups. We definitely plan to study these interesting extensions.
>
> **When evaluating, accuracy seems just an indicator and not really a measure of distributional discrepancy like mmd and adv. Do I understand it right?**
>
> Yes, we report the discrepancy measures in Table 2 but included accuracy as an evaluation criterion simply to check that the representation still retains enough signal to yield good performance for the regression/classification task at hand. We welcome suggestions on including any other relevant measures. Thank you.
>
> We appreciate the questions and the observations. Please let us know if we have satisfactorily answered the questions.

---

### Official Review · Reviewer_KLWx · 2023-10-31

**Soundness:** 3 good
**Presentation:** 3 good
**Contribution:** 3 good
**Rating:** 8
**Confidence:** 4

**Summary:**

Authors present a Category Theory-based formalism for covariate shifts
and a method derived from this formalism for pooling data with
covariate shifts for simple tasks. Images are mapped to a latent space
- using AE but I guess this can be other things - and covariate shifts
are modeled as transformations in the latent space. Invariance to
some covariate shifts, e.g., using different acquisition devices, are
modeled by enforcing similar latent space
representations. Equivariance is obtained by enforcing a chosen
transformation. Experimens show that the resulting model can retain
accuracy while removing differences between samples with covariate
shift in the latent space in cases where invariance is
desired. When equivariance is desired, authors show that the latent
space transformations are also good.

**Strengths:**

1. The general formalism Category Theory provides is elegant and
   contains a class of algorithms as special cases. This has the
   potential of inventing new directions for data pooling and
   addressing covariate shifts.
2. Authors provide a great introduction to the theory and maps the
   relevant aspects to practical problems pertinent to covariate
   shifts observed in application in neuroimaging.
3. Experimental results are motivating. Authors compare with some
   recent work to convince us that the model proposed is indeed
   useful.
4. The article is not necessarily providing a novel technique but
   rather discusses a theory that views already applied methods as a
   special case and thus allows further generalization. For instance,
   https://link.springer.com/chapter/10.1007/978-3-031-16431-6_1
   applied similar ideas for longitudinal imaging data.

**Weaknesses:**

1. The current method requires overlapping covariate values between
   data sets coming from different centers. Equation (9) seems crucial
   for the methods to work. However, until (9) the reader is left with
   the opinion that Category Theory would magically work for
   non-overlapping covariate values between different data sets. This
   should be clearly discussed early on in the article to avoid any
   confusions and - to be honest - false hope. Given two data sets
   whose covariate values do not overlap, the current method may not
   work at all.
2. The emphasis on couterfactuals is rather questionable. There is no
   decoupling of the effects of different covariates on data. In other
   words, a transformation in the latent space may correspond to
   changes in multiple covariates. A well curated data set may avoid
   this problem, however, generally this will not hold. Therefore, it
   is difficult to discuss about counterfactuals and even
   interventions. The latent space transformations will not be able to
   account a missing causal model linking covariates, observed
   images and labels. I recommend authors to reconsider claims
   regarding counterfactuals.
3. Figure 5 is not explained at all. This seems like an important
   figure and requires further explanation in the text.
4. I find the claim that this approach simplifies recent
   works in the literature not substantiated. Furthermore, without
   explicit disentanglement nor the capability to do interventions, I
   am not sure to what authors may be referring.

**Questions:**

1. I encourage authors to reconsider their claims, especially
   concerning counterfactuals and models requirements of overlapping
   covariate values between different sets.

---

> ### Author Response · Authors · 2023-11-22
>
> Dear Reviewer, We thank you for the constructive and thoughtful feedback. Your comments have helped us refine the paper’s message in the revised version we uploaded. We are grateful to see appreciation of the main formulation, and answer all questions below.
>
> **The current method requires overlapping covariate values between data sets coming from different centers.**
>
> We are sorry that the text does not make this explicit. We have now modified it so that there is no confusion. We assumed that Figure 1 would convey the setup accurately but we have now annotated it more. The figure shows three covariates: they do not match exactly but do have a shared interval. The problem setup is also described in the introduction (middle of page 2). We note that we deal with “**some** shift/disbalance in covariates across sites (participant data including age, sex and so on, which influence the scans)”. But yes, to avoid any confusion, we made the text much more explicit. Thank you for pointing this out.
>
> We agree that if there is no overlap between the covariates, then there is no coupling to work with, and it is not clear how to write down a sensible formulation.
>
> **The emphasis on counterfactuals is rather questionable**
>
> Yes, we agree completely. The use of the term counterfactuals was simply to draw an analogy between our “what if” experiment to a topic being actively studied by many in the community. We value the constructive nature of this comment. We have removed this terminology to avoid any confusion. Thanks.
>
> **Figure 5 is not explained at all. This seems like an important figure and requires further explanation in the text.**
>
> We included an explanation of Figure 5 at the bottom of page 8. But based on the suggestion, we have expanded the passage with more information and intuition about the figure. Roughly, the idea is that if the model has learned a meaningful structure, then when applying a morphism that increases the covariate pertaining to a risk factor, the new vector should be much closer to the diseased group (high cosine similarity and/or smaller distance).
>
> **I find the claim that this approach simplifies recent works in the literature not substantiated.**
>
> A recent baseline (see Table 1) uses a multi-stage pipeline for training, and handles one continuous and one categorical covariate. We hope that the reviewer agrees that Alg. 1 is much more concise and simple, and is suggested directly by the category-theoretic casting of the problem. We agree that disentanglement or mutual information-based terms as in Moyer et al (references on top of page 13) can be added to our model.
>
>
> We appreciate the suggestions, particularly regarding counterfactuals and have updated the paper. Please let us know if we have satisfactorily answered the questions.

---

> > ### Comment · Reviewer_KLWx · 2023-11-22
> > **thank you**
> >
> > I appreciate authors' responses and modifications applied to the article.
> > I believe that the presented view is concise and may lead to lively and constructive discussions in the venue.

---

> > > ### Author Response · Authors · 2023-11-22
> > > **Thank you**
> > >
> > > We appreciate the time and effort of the reviewer. We are thankful for the strong vote of confidence.

---

### Author Response · Authors · 2023-11-22
**General response**

Thanks to all the reviewers for appreciating the strengths of the paper and for suggesting various improvements. We answer all questions/clarifications in the review below. We have also revised and uploaded the paper and we welcome any feedback or questions.

---

### Meta-Review · Area_Chair_LRra · 2023-12-08

**Metareview:**

This research addresses the challenge of handling shifts and imbalances in covariates, particularly secondary non-imaging data, when pooling small, similar datasets across multiple sites or institutions. This is common in disciplines where it's necessary to study weak but relevant associations between images and disease incidence. While these issues have been studied for classical models, the solutions do not directly apply to overparameterized Deep Neural Network (DNN) models.

Recent work has shown that strategies from fairness and invariant representation learning provide a meaningful starting point, but current methods can only account for shifts/imbalances in a few covariates at a time. The authors present a solution to this problem by viewing it through the lens of Category theory, offering a simple and effective solution that avoids the need for elaborate multi-stage training pipelines.

The authors demonstrate the effectiveness of their approach through extensive experiments on real datasets. Furthermore, they discuss how their formulation provides a unified perspective on over five distinct problem settings in vision, from self-supervised learning to matching problems in 3D reconstruction.

In summary, this research offers a novel approach to handling shifts and imbalances in covariates in deep learning models by leveraging Category theory. This allows for more effective handling of small, similar datasets across multiple locations, with potential applications in multiple vision-related problem settings.

**Justification For Why Not Higher Score:**

The introduction of category theory is novel and interesting to the society, the acceptance could be bumped up toward spotlight. Yet majority of reviewers suggests acceptance slightly above the borderline, so is the final decision.

**Justification For Why Not Lower Score:**

There is one reviewer who rated this paper slightly below the borderline with minor confidence (1), but did not give any specific comments on the weakness or questions. So this review is ignored.

---

### Decision · Program_Chairs · 2024-01-16

Accept (poster)